

# Lectins: an effective tool for screening of potential cancer biomarkers

Onn Haji Hashim[1,2], Jaime Jacqueline Jayapalan[2] and Cheng-Siang Lee[1]

[1] Department of Molecular Medicine, Faculty of Medicine, University of Malaya, Kuala Lumpur, Malaysia
[2] University of Malaya Centre for Proteomics Research, Faculty of Medicine, University of Malaya, Kuala Lumpur, Malaysia

## ABSTRACT

In recent years, the use of lectins for screening of potential biomarkers has gained increased importance in cancer research, given the development in glycobiology that highlights altered structural changes of glycans in cancer associated processes. Lectins, having the properties of recognizing specific carbohydrate moieties of glycoconjugates, have become an effective tool for detection of new cancer biomarkers in complex bodily fluids and tissues. The specificity of lectins provides an added advantage of selecting peptides that are differently glycosylated and aberrantly expressed in cancer patients, many of which are not possibly detected using conventional methods because of their low abundance in bodily fluids. When coupled with mass spectrometry, research utilizing lectins, which are mainly from plants and fungi, has led to identification of numerous potential cancer biomarkers that may be used in the future. This article reviews lectin-based methods that are commonly adopted in cancer biomarker discovery research.

## BIOLOGY OF LECTINS

Lectins are carbohydrate binding proteins which are found ubiquitously in nature. The term 'lectin' originates from the Latin word *legere,* which means to choose or to select (*Boyd & Shapleigh, 1954*). By binding to carbohydrates, lectins serve diverse biological functions. Plant lectins, which typically cause agglutination of certain animal cells, play important roles in defense against invasion of virus, bacteria or fungi (*Dias et al., 2015*). They are also believed to mediate symbiosis relationship between plants and microorganisms (*De Hoff, Brill & Hirsch, 2009*), and some may be involved in regulatory and signaling pathways in plant cells (*Chen et al., 2002*).

Lectins have initially been classified based on their binding to different glycan structures. They were categorized either as galactose, *N*-acetylglucosamine (GlcNAc), *N*-acetylgalactosamine (GalNAc), glucose, L-fucose, mannose, maltose, sialic acid-specific or complex glycan-binding lectins (*Lis & Sharon, 1986*). Later, they were also classified based on the characteristics and numbers of their carbohydrate binding domains, namely merolectins, hololectins, chimerolectins and superlectins (*Peumans et al., 2001*). With the emergence of detailed structural properties of lectins being elucidated via the advancement of technology, this classification further evolved into that based on distinct protein folding,

Corresponding author
Onn Haji Hashim,
onnhashim@um.edu.my

**Table 1** Summary of different applications of lectins in medical research and therapy.

| Lectin applications | Reference |
| --- | --- |
| Antibacterial agent | *Saha et al. (2014)*, *Dias et al. (2015)* |
| Antifungal agent | *Klafke et al. (2013)*, *Regente et al. (2014)* |
| Antiparasitic agent | *Tobata-Kudo, Kudo & Tada (2005)*, *Heim et al. (2015)* |
| Antiviral agent | *Lusvarghi & Bewley (2016)*, *Monteiro & Lepenies (2017)* |
| Biomarker for disease detection and monitoring | This review article |
| Drug delivery | *Leong et al. (2011)*, *Neutsch et al. (2013)* |
| Induction of immunological and inflammatory response | *Singh et al. (2011)*, *Ditamo et al. (2016)* |
| Inhibition of cancer cell adhesion | *Redondo & Alvarez-Pellitero (2010)*, *Silva et al. (2014)* |
| Inhibition of cancer cell growth/antitumor agent | *Jebali et al. (2014)*, *Quiroga, Barrio & Añón (2015)* |
| Promotion of healing in cutaneous wounds | *Brustein et al. (2012)*, *Coriolano et al. (2014)* |

domains/structural similarities and evolutionary-relatedness of proteins (*Peumans et al., 2001*). Via this categorization, 12 different lectin families, which include *Agaricus bisporus* agglutinin homologues, amaranthins, class V chitinase homologues with lectin activity, cyanovirin family, *Euonymus europaeus* agglutinin family, *Galanthus nivalis* agglutinin family, jacalins, lysin motif domain, nictaba family, proteins with hevein domains, proteins with legume lectin domains and ricin-B family (*Van Damme, Lannoo & Peumans, 2008*), have been derived.

Ricin is believed to be the first lectin discovered in the seeds of the castor bean plant, *Ricinus communis*, in 1888 (*Sharon & Lis, 2004*). Paradoxically, research on lectin only flourished several decades subsequent to ricin's discovery after James Sumner successfully purified a crystalline protein from jack bean (*Canavalia ensiformis*) in 1919. Sumner later showed that the protein caused agglutination of cells such as erythrocytes and yeast. The agglutinin, which is now known as concanavalin A or ConA, was also used for the first time to demonstrate binding of lectins to carbohydrate. To date, there are more than a thousand plant species that have been reported to possess lectins. Most of these lectins are in abundance in seeds (*Lis & Sharon, 1986*; *Benedito et al., 2008*), whilst some are found in leaves, roots, flower, sap, barks, rhizomes, bulbs, tubers and stems (*Dias et al., 2015*). Because of their carbohydrate binding specificities, many lectins have been increasingly applied in different areas of medical research and therapy (Table 1).

## CANCER BIOMARKER

A biomarker is defined as "a characteristic that is objectively measured and evaluated as an indicator of normal biological processes, pathogenic processes or pharmacologic responses to a therapeutic intervention" (*Biomarkers Definition Working Group, 2001*). Hence, simple parameters from pulse and blood pressure to protein constituents of cells, tissues, blood and other biofluids are classified as biomarkers. Bodily fluids that have been mined for cancer biomarkers thus far include serum/plasma, urine, saliva and other tissue-specific fluids

such as seminal fluid, cerebrospinal fluid, bone marrow aspirates, etc. Cancer biomarkers are useful for early detection, diagnosis and prognosis of the disease. They are also heavily relied on in management of patients, and assessment of pharmacodynamics of drugs, risk, as well as recurrence of the disease.

Efforts in the search for new cancer biomarkers remain active even in the present day. Currently, there are only a handful of cancer biomarkers that are commonly being used in the clinical setting (Table 2), most of which have been officially approved by the US Food and Drug Administration (FDA) for clinical use (*Füzéry et al., 2013*). More are definitely needed for improved detection and diagnosis, particularly when the reliability of many of the FDA approved biomarkers remain a problem due to their limited levels of sensitivity and specificity. For example, CA-125 which is used as a biomarker for ovarian cancer, is also often elevated in other cancers such as those of the breast (*Norum, Erikstein & Nustad, 2001*), lung (*Salgia et al., 2001*) and colon or rectum (*Thomas et al., 2015*). Similarly, prostate specific antigen (PSA), a tissue-specific serum protein that is used in the diagnosis of prostate cancer, is also commonly increased in sera of patients with benign prostatic hyperplasia, thus, posing difficulties in clinically differentiating the two different conditions (*Barry, 2001*; *Thompson et al., 2004*). These limitations, together with the recent development of various state-of-the-art methodologies including genomics, proteomics and bioinformatics, have consequentially propelled research towards identification of new cancer biomarkers that are more sensitive and specific.

Amongst bodily fluids that have been mined for cancer biomarkers, serum/plasma is most popular. Serum or plasma has the advantage of being routinely sampled in clinical investigations. However, the extreme complexity and broad dynamic range of protein abundance in serum and plasma pose a formidable challenge in research screening for potential cancer biomarkers, which mostly comprise low abundance glycoproteins. Because of this, many cancer biomarker exploratory studies involving serum or plasma often involved enrichment and/or pre-fractionation of the samples using techniques such as immunodepletion (*Prieto et al., 2014*), immunoprecipitation (*Lin et al., 2013*) and size-exclusion chromatography (*Hong, Koza & Bouvier, 2012*). However, the use of such techniques, despite their wide applications in biomarker discovery investigations, is generally unable to make a significant difference in unmasking proteins of low abundance (*Polaskova et al., 2010*), and may result in concomitant loss of non-targeted proteins (*Bellei et al., 2011*).

## APPLICATIONS OF LECTINS IN CANCER BIOMARKER DISCOVERY RESEARCH

Interestingly, the majority of cancer biomarkers that are currently being used in the clinical settings are glycoproteins, which are structurally altered in their glycan moieties and aberrantly expressed (*Henry & Hayes, 2012*). However, only alpha-fetoprotein (AFP) and CA15-3 are clinically monitored for their glycan changes in the therapy for hepatocellular carcinoma and breast cancer, respectively. The other cancer biomarkers are being monitored for their total protein levels (*Kuzmanov, Kosanam & Diamandis, 2013*). Indeed, changes in glycosylation are believed to be a main feature in oncogenic transformation
**Table 2** List of commonly used tumor markers in clinical practice.

| Biomarker | Glycosylated | Cancer type | Specimen | Clinical use |
|---|---|---|---|---|
| Alpha-feto protein (AFP) | Yes | Testicular | Serum/plasma; Amniotic fluid[a] | Management of cancer |
| AFP-L3% | Yes | Hepatocellular | Serum | Risk assessment |
| Beta-2-microglobulin (B2M) | Yes | Blood cells | Serum, Urine, Cerebrospinal fluid | Monitoring progression and recurrence |
| Bladder tumor-associated antigen | Unknown | Bladder | Urine | Monitoring disease |
| CA 15–3 | Yes | Breast | Serum/plasma | Monitoring disease; Response to therapy |
| CA 19–9 | Yes[b] | Pancreatic | Serum/plasma | Monitoring disease |
| CA 27–29 | Yes | Breast | Serum | Monitoring disease; Response to therapy |
| CA 125 | Yes | Ovarian | Serum/plasma | Monitoring disease; Response to therapy |
| Carcinoembryonic antigen (CEA) | Yes | Colon | Serum/plasma | Monitoring disease; Response to therapy |
| c-Kit | Yes | Gastrointestinal stromal tumors | Tissue | Detection of tumor; Patient selection |
| EpCAM, CD45, cytokeratins 8, 18+, 19+ | Yes | Breast | Whole blood | Monitoring progression and survival |
| Epidermal growth factor receptor (EGFR) | Yes | Colon | Tissue | Therapy selection |
| Estrogen receptor (ER) | Yes | Breast | Tissue | Prognosis; Response to therapy |
| HER2/NEU | Yes | Breast | Serum; Tissue | Monitoring progression; Therapy selection |
| Human chorionic gonadotropin | Yes | Testicular | Serum | Staging of cancer |
| Human epididymis protein 4 (HE4) | Yes | Ovarian | Serum | Monitoring progression and recurrence |
| Fecal occult blood (haemoglobin) | Yes | Colorectal | Feces | Detection of tumor |
| Fibrin/fibrinogen degradation product (DR-70) | Yes | Colorectal | Serum | Monitoring disease |
| Free prostate specific antigen | Yes | Prostate | Serum | Screening for disease |
| Nuclear mitotic apparatus protein (NuMA, NMP22) | Yes | Bladder | Urine | Diagnosis and monitoring disease |
| p63 protein | No | Prostate | Tissue | Differential diagnosis |
| Plasminogen activator inhibitor (PAI-1) | Yes | Breast | Tissue | Monitoring disease; Therapy selection |
| Progesterone receptor (PR) | Yes | Breast | Tissue | Therapy selection |
| Pro2PSA | Yes | Prostate | Serum | Discriminating cancer from benign disease |
| Thyroglobulin (Tg) | Yes | Thyroid | Serum/plasma | Monitoring disease |
| Total PSA | Yes | Prostate | Serum | Diagnosis and monitoring disease |
| Urokinase plasminogen activator (uPA) | Yes | Breast | Tissue | Monitoring disease; Therapy selection |

**Notes.**
[a] Also used in prenatal diagnosis of birth defects, a non-cancer application.
[b] A tetrasaccharide carbohydrate that is usually attached to O-glycans on the surface of cells.

as glycans are known to be continuously involved in cancer evolving processes, such as cell signaling, angiogenesis, cell–matrix interactions, immune modulation, tumor cell dissociation and metastasis. Glycosylation changes that are commonly associated with cancer transformation include sialylation, fucosylation, increased GlcNAc-branching of N-glycans, and overexpression of truncated mucin-type O-glycans (*Pinho & Reis, 2015*). Hence, it is not surprising that lectin-based approaches are becoming more popular in

studies screening for novel cancer biomarkers. Table 3 shows a list of lectins that have been used in cancer biomarker discovery research. In the following sections of this review, the applications of lectins in cancer biomarker discovery, including immobilized lectin affinity chromatography, enzyme-linked lectin assay, lectin histochemistry, lectin blotting and lectin array, are addressed. For lectin-based biosensor analysis, readers are recommended to refer to separate review articles (*Pihíková, Kasák & Tkac, 2015*; *Coelho et al., 2017*).

## IMMOBILIZED-LECTIN AFFINITY CHROMATOGRAPHY

Immobilized-lectin affinity chromatography is a method for separation of glycoproteins based on a highly specific interaction between a lectin, which is immobilized onto a chosen matrix, and its carbohydrate ligands (*Hage et al., 2012*). The technique, when complemented with mass spectrometry analysis, provides a useful tool in research aiming to identify potential cancer biomarkers (Fig. 1). By comparing bodily fluid samples of control subjects with those from patients with cancer, glycoproteins that are aberrantly expressed or differently glycosylated from the resulting glycoprotein-enriched eluates can be easily identified. Immobilized-lectin affinity chromatography is currently one of the most widely employed techniques for enrichment of glycoproteins in cancer biomarker research.

By using immobilized-ConA, followed by separation by 2-dimensional gel electrophoresis (2-DE), *Rodriguez-Pineiro et al. (2004)* were able to profile serum samples of patients with colorectal cancer and showed significant altered expression of several *N*-glycosylated proteins that were identified by mass spectrometry. These included up-regulated expression of haptoglobin and lowered expression of antithrombin-III, clusterin, inter-alpha-trypsin inhibitor heavy chain H4, beta-2-glycoprotein I and coagulation factor XIII B chain in the colorectal cancer patients relative to healthy donors. Similarly, *Seriramalu et al. (2010)* reported the lowered expression of complement factor B and alpha-2 macroglobulin in patients with nasopharyngeal carcinoma relative to controls using the champedak mannose binding lectin. In the case of *O*-glycosylated proteins, considerable studies have been reported using champedak galactose binding (CGB) lectin, which has a unique characteristic of binding to the *O*-glycan structures of glycoproteins (*Abdul Rahman et al., 2002*) in serum and urine samples. Cancers that have been investigated using immobilized-CGB lectin include endometrial cancer (*Mohamed et al., 2008*) and prostate cancer (*Jayapalan et al., 2012*). However, most of the serum and urine *N*- and *O*-glycosylated proteins that were isolated using the immobilized-lectin affinity chromatography are not directly cancer associated but the body's highly abundant acute-phase reactant proteins (*Pang et al., 2010*).

More recently, analyses of enriched glycopeptide eluates of immobilized-lectin affinity chromatography for identification of site-specific glycosylation using mass spectrometry techniques have been reported in studies in search of potential cancer biomarkers. Enrichment of core fucosylated peptides using *Lens culinaris* agglutinin (LCA) after trypsin digestion of glycoproteins, followed by endo F3 partial deglycosylation and nano LC-MS/MS methodologies, has led to identification of glycopeptides that can potentially be

**Table 3 List of lectins used in cancer biomarker discovery research.**

| Lectin | Abbreviation | Specificity | Glycan linkage | References |
|---|---|---|---|---|
| African legume (*Griffonia (Bandeiraea) simplicifolia*) lectin-I | GSLI (BSLI) | α-Gal; α-GalNAc | O-linked | *Lescar et al. (2002)* |
| Asparagus pea (*Lotus tetragonolobus*) lectin | LTL | Fucα1-3(Galβ1-4)GlcNAc, Fucα1-2Galβ1-4GlcNAc | N-linked | *Pereira & Kabat (1974)*, *Yan et al. (1997)* |
| Koji (*Aspergillus oryzae*) lectin | AOL | α1,6-fucosylated | N-linked | *Matsumura et al. (2007)* |
| Castorbean (*Ricinus communis*) agglutinin | RCA | Galβ1-4GlcNAc; terminal β-D-Gal | N-linked | *Harley & Beevers (1986)*, *Wang et al. (2011)* |
| Champedak (*Artocarpus integer*) galactose binding lectin | CGB | Gal; GalNAc | O-linked | *Hashim et al. (1991)*, *Gabrielsen et al. (2014)* |
| Champedak (*Artocarpus integer*) mannose binding lectin | CMB | Man | N-linked | *Lim, Chua & Hashim (1997)*, *Gabrielsen et al. (2014)* |
| Daffodil (*Narcissus pseudonarcissus*) lectin | NPL | α-Man, prefers polymannose structures containing α-1,6 linkages | N-linked | *Kaku et al. (1990)*, *Lopez et al. (2002)* |
| Elderberry (*Sambucus nigra*) agglutinin | SNA | Neu5Acα2-6Gal(NAc)-R | N- and O-linked | *Shibuya et al. (1987)*, *Silva, Gomes & Garcia (2017)* |
| Gorse or furze (*Ulex europaeus*) seed agglutinin-I | UEA-I | Fucα1-2Gal-R | N- and O-linked | *Holthofer et al. (1982)*, *Rudrappan & Veeran (2016)* |
| Jackbean (*Canavalia ensiformis*) lectin | ConA | α-Man; α-Glc | N-linked | *Percin et al. (2012)* |
| Jackfruit (*Artocarpus heterophyllus*) lectin | Jacalin | Gal; GalNAc | O-linked | *Kabir (1995)*, *Jagtap & Bapat (2010)* |
| Lentil (*Lens culinaris*) hemagglutinin | LcH | Man; Glc (Affinity enhanced with α-Fuc attached to N-acetylchitobiose) | N-linked | *Howard et al. (1971)*, *Chan et al. (2015)* |
| Amur maackia (*Maackia amurensis*) lectin II | MAL II | Siaα2-3Galβ1-4GlcNAc; Siaα2-3Galβ1-3GalNAc | N- and O-linked | *Konami et al. (1994)*, *Geisler & Jarvis (2011)* |
| Orange peel fungus (*Aleuria aurantia*) lectin | AAL | Fucα1-6GlcNAc; Fucα1-3LacNAc | N- and O-linked | *Hassan et al. (2015)* |
| Peanut (*Arachis hypogaea*) agglutinin | PNA | Galβ1-3GalNAc; Gal | O-linked | *Chacko & Appukuttan (2001)*, *Vijayan (2007)* |
| Chinese green dragon (*Pinellia pedatisecta*) agglutinin | PPA | Man | N-linked | *Li et al. (2014)* |
| Poke weed (*Phytolacca americana*) mitogen lectin | PWM | GlcNAc oligomers | N-linked | *Kino et al. (1995)*, *Ahmad et al. (2009)* |
| Red kidney bean (*Phaseolus vulgaris*) lectin | PHA-L | Bisecting GlcNAc | N-linked | *Kaneda et al. (2002)*, *Movafagh et al. (2013)* |
| Thorn-apple (*Datura stramonium*) lectin | DSL | (GlcNAcβ4)n | N-linked | *Yamashita et al. (1987)*, *Abbott et al. (2010)* |
| Wheat germ (*Triticum vulgaris*) agglutinin | WGA | GlcNAcβ1-4GlcNAc β1-4GlcNAc; Neu5Ac | N-linked | *Nagata & Burger (1972)*, *Parasuraman et al. (2014)* |
| White button mushroom (*Agaricus bisporus*) lectin | ABL | GalNAc; Galβ1,3GalNAc (T antigen); sialyl-Galβ | O-linked | *Nakamura-Tsuruta et al. (2006)*, *Hassan et al. (2015)* |
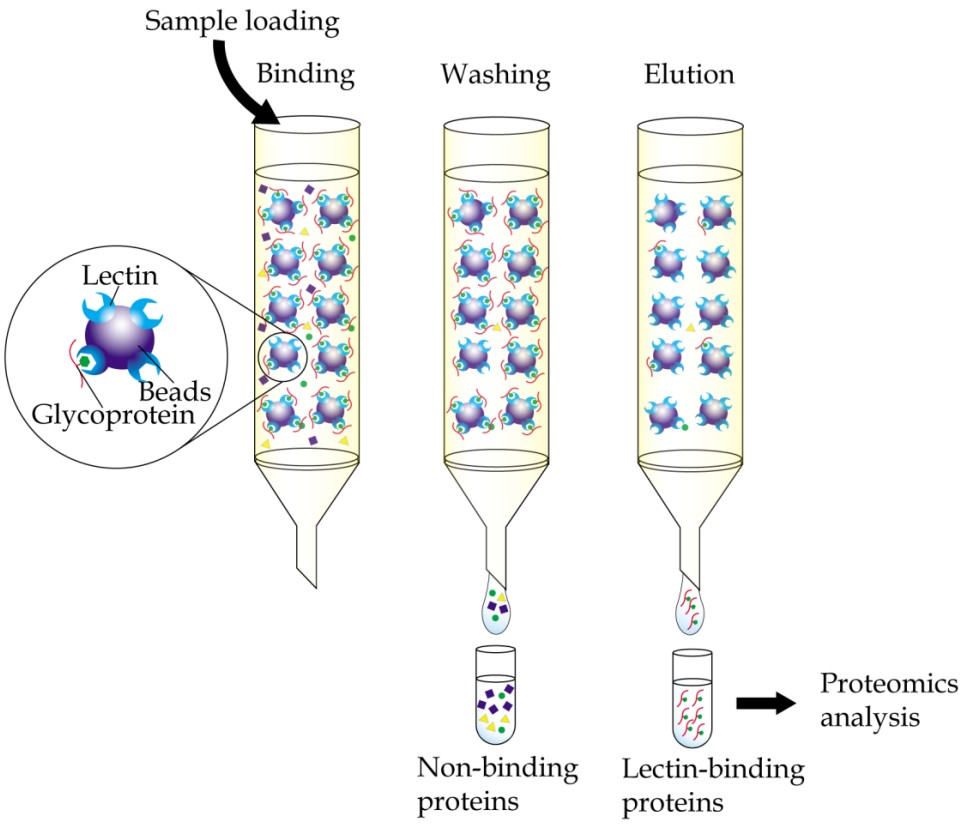

**Figure 1** **General workflow of immobilized-lectin affinity chromatography.** Bodily fluid of cancer patients can be assayed for potential cancer biomarkers by running it through a chromatography column packed with a gel matrix that is conjugated with a lectin of interest. Non-binding proteins are then washed out, whilst bound glycoproteins are eluted using specific carbohydrate solutions. The lectin bound glycoproteins are finally identified using proteomics analysis.

used as diagnostic biomarkers for pancreatic cancer (*Tan et al., 2015*). Similarly, enrichment of trypsin-digested glycopeptides using *Aleuria aurantia* lectin (AAL) that was immobilized onto agarose gel, followed by analysis using LC/MS, has resulted in identification of alpha-1-acid glycoprotein with multi-fucosylated tetraantennary glycans as a potential marker for hepatocellular carcinoma (*Tanabe et al., 2016*). In another study, the *Sambucus niagra* agglutinin (SNA) affinity column was used to separate various glycoforms of serum PSA according to the types of sialic acid linkages (*Llop et al., 2016*). This has resulted in identification of $\alpha2, 3$-sialylated PSA as a marker for discriminating patients with high-risk prostate cancer from those with benign prostatic hyperplasia and low-risk prostate cancer, with higher levels of sensitivity and specificity.

Another variant of immobilized-lectin affinity chromatography used in cancer biomarker research is multi-lectin affinity chromatography. Since no single lectin is able to isolate the complete complement of a glycoprotein, a multi-lectin affinity chromatography is gaining popularity because of its greater coverage and depth of analyses. Using a combination of four different types of lectins, including ConA, SNA, *Phaseolus vulgaris* agglutinin (PHA) and *Ulex europaeus* agglutinin (UEA), for sequential multi-lectin affinity

chromatography in silica-based microcolumns and nano-LC/MS/MS for identification of proteins, *Madera et al. (2007)* successfully profiled glycoproteins from microliter volumes of serum. Along the same line but using ConA, wheat germ agglutinin (WGA) and jacalin that were integrated into an automated HPLC platform and immuno-depleted serum samples, *Zeng et al. (2011)* demonstrated a comprehensive detection and changes in the abundances of post-translationally modified breast cancer-associated glycoproteins. To facilitate a cascading flow of samples from column to column for simultaneous and efficient capturing and enrichment of fucosylated proteins, *Selvaraju & EI Rassi (2013)* developed of a platform, which comprised multi-lectin columns driven by HPLC pumps for elucidating differential expression of serum fucome between cancer-free and breast cancer subjects. This method surpasses issues such as loss of samples due to sample preparation and processing (e.g., dilution) as well as other experimental biases that commonly occur when using other techniques.

Recently, *Miyamoto et al. (2016)* reported a comprehensive proteomic profiling of ascites fluid obtained from patients with metastatic ovarian cancer enriched by differential binding to multiple lectins, including ConA, AAL and WGA. Alpha-1-antichymotrypsin, alpha-1-antitrypsin, ceruloplasmin, fibulin, fibronectin, hemopexin, haptoglobin and lumican appeared more abundant in ascites of the patients compared to controls. Further glycopeptide analysis identified unusual *N*- and *O*-glycans in clusterin, fibulin and hemopexin glycopeptides, which may be important in metastasis of ovarian cancer. Similar use of multi-lectin affinity chromatography for enrichment of *N*-linked glycoproteins by *Qi et al. (2014)* has successfully identified human liver haptoglobin, carboxylesterase 1 and procathepsin D as candidate biomarkers associated with development and progression of hepatocellular carcinoma. Whilst the concentrations of human liver haptoglobin and carboxylesterase 1 were consistently lower, higher concentration of procathepsin D was detected in the liver cancer tissues. Further in-depth analysis projected the promising use of procathepsin D as a serological biomarker for diagnosis of hepatocellular carcinoma.

## ENZYME-LINKED LECTIN ASSAY

Enzyme-linked lectin assay is a method that adopts the principle of enzyme-linked immunosorbent assay but uses lectin as one of the reagents instead of antibody. This method was introduced by *McCoy Jr, Varani & Goldstein (1983)* in the early eighties. In a direct assay, samples that contain glycoconjugates may be coated directly onto the wells of a microtiter plate, followed by addition of an enzyme-conjugated lectin, which will then bind to their glycan structures (Fig. 2A). The enzyme converts a colorless substrate solution to a colored product, that is then measured using a spectrophotometer, and whose intensity is used to estimate the levels of the coated glycoconjugates. Depending on the structures of glycans that need to be detected, specific lectins are carefully selected. The enzyme-linked lectin assay has been used in a plethora of research including those of cancer biomarkers (*Kuzmanov, Kosanam & Diamandis, 2013*). It is easy to perform, very cost effective and requires minute amounts of samples. One drawback of the direct enzyme-linked lectin assay is that glycoproteins that are detected may not be identifiable unless it is coupled with proteomics analysis or antibody detection.

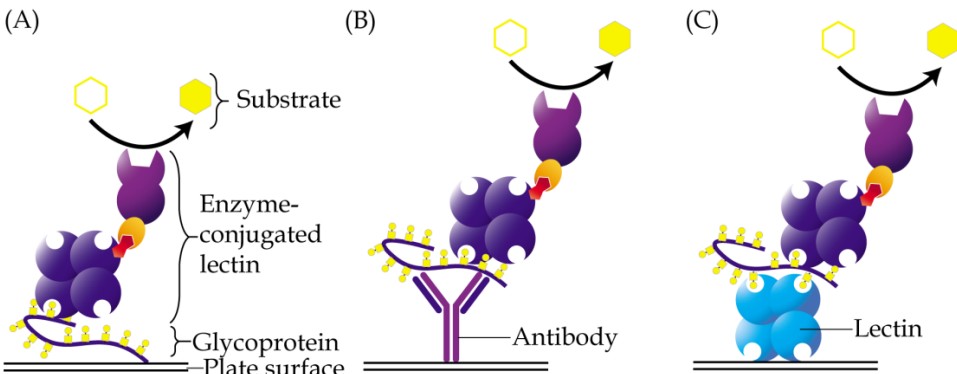

**Figure 2 Different approaches of enzyme-linked lectin assay.** (A) In the direct assay, coating of samples is performed directly onto the surface of a microtiter plate, followed by addition of enzyme-conjugated lectin. (B) In the hybrid assay, antibody is instead coated onto the plate to capture specific glycoproteins of interest, prior to addition of the enzyme-conjugated lectin. (C) Sandwich enzyme-linked lectin assay is an alternative method involving two different lectins. The first lectin is coated onto plates and used as a capturing reagent, whilst the second lectin is used as detection reagent. For all the aforementioned methods, glycoproteins are usually detected using a lectin that is conjugated to an enzyme, which then converts a specific substrate into a colored product.

Based on their earlier study that identified a predominantly high molecular weight glycoprotein that binds to peanut lectin (PNA) in the sera of patients with pancreatic cancer, *Ching & Rhodes (1989)* developed a direct enzyme-linked PNA assay for diagnosis of pancreatic cancer. Results obtained from the lectin-based assay were apparently found to be comparable with those derived from using CA19-9 radioimmunoassay in terms of sensitivity and specificity for pancreatic cancer. In another study, *Reddi et al. (2000)* reported the use of similar enzyme-linked PNA assay to estimate the levels of Thomsen-Friendenreich antigen (T-Ag) in sera of patients with squamous cell carcinoma of the uterine cervix, before and after radiotherapy. The study demonstrated significantly higher levels of T-Ag in the sera of the uterine cervical cancer patients compared to normal individuals, and that the expression of PNA-binding T-Ag were directly proportional to the aggressiveness of the cancer. In a study by *Dwek, Jenks & Leathem (2010)*, the specificity of UEA-1 lectin to α1,2-linked fucose sites was capitalized for detection of fucosylated serum free PSA in a direct enzyme-linked lectin assay. Their results demonstrated higher levels of fucosylated serum free PSA in patients with prostate cancer compared to those with benign prostatic hyperplasia.

Aside from sera, the direct enzyme-linked lectin assay has also been used in the analysis of tissue lysate glycoproteins. In a recent study of breast cancer tissue lysates of different stages, *Wi et al. (2016)* demonstrated increased interaction with ConA, *Ricinus communis* Agglutinin I, AAL and *Maackia amurensis* lectin II (MAL II) relative to normal tissue specimen of the same subjects. This is generally interpreted to show enhanced mannosylation, galactosylation, sialylation and fucosylation of glycoproteins in the breast cancer tissues. In another study, *Kim et al. (2014)* have shown lower levels of fucosylation and sialylation of cytosolic intracellular glycoproteins in cancerous human cervical tissues compared to normal tissue specimens from the same subjects using AAL and SNA lectins,

respectively. However, the levels of mannosylation, which was assayed using ConA, were not significantly different between cancer tissues and normal specimens.

Subtle changes to the classical enzyme-linked lectin assay protocol have been introduced over the years. An example is the combined use of antibody with lectin to enable detection of glycosylation on a specific protein (*Kim, Lee & Kim, 2008*). In this case, an antibody may be coated directly onto the wells of a microtiter plate, which will allow pre-capturing of a protein of interest from complex samples (Fig. 2B). A lectin is then added and let on to bind with the glycan structures of the protein. In this method, prior purification of a glycoprotein is not needed as the antibody utilized specifically isolates the protein of interest from within the samples. This method is also more suitable for glycoprotein antigens, which are generally hydrophilic and cannot be well-coated onto a microtiter plate. The disadvantage of this approach is that a lectin may directly interact with glycan chains of the antibody used, which would then result in high background readings.

To solve the issue of the non-specific direct interaction of lectin to antibodies in enzyme-linked lectin assays, *Takeda et al. (2012)* have instead used the Fab fragment of anti-human haptoglobin IgG antibody and biotinylated AAL lectin for sandwich detection of fucosylated haptoglobin. Their results showed that the levels fucosylated haptoglobin were significantly associated with overall and relapse-free survival, distant metastasis, clinical stage, and curability of patients with colorectal cancer. When Kaplan–Meier analysis was performed on patients after more than 60 months of surgery, positive cases of fucosylated-haptoglobin showed poor prognosis compared with fucosylated-haptoglobin negative cases. This leads to the suggestion of fucosylated haptoglobin as a prognostic marker in addition to CEA for colorectal cancer. Along the same line, *Jin et al. (2016)* have instead used protein A as the capturing reagent and AAL lectin as detection probe, for assessment of fucosylated circulating antibodies in cervical intraepithelial neoplasia and cervical cancer. Significantly lower levels of fucosylated circulating immunoglobulins were shown in female patients with cervical cancer compared to those with cervical intraepithelial neoplasia or normal subjects.

In a reverse contrast strategy, *Wu et al. (2013)* have used SNA lectin to capture sialylated glycoproteins and biotinylated-antibodies to detect clusterin, complement factor H, hemopexin and vitamin D-binding protein to validate the altered levels of the respective glycoproteins in sera of patients with ovarian cancer. The results were consistent with their data that was previously generated using isobaric chemical labeling quantitative strategy. In a similar strategy, *Liang et al. (2015)* have used *Bandeiraea (Griffonia) simplicifolia*-I (BSI), AAL and Poke weed mitogen (PWM) lectins as capturing reagents and biotinylated anti-human α-1-antitrypsin polyclonal antibody in a sandwich enzyme-linked lectin combination assay to validate results of their lectin microarray analysis of serum samples of patients with lung cancer. While galactosylated α-1-antitrypsin was shown to demonstrate remarkable discriminating capabilities to differentiate patients with non-small-cell lung cancer from benign pulmonary diseases, their fucose- and poly-LacNAc-containing counterparts may be used to discriminate lung adenocarcinoma from benign diseases or other lung cancer subtypes, and small-cell lung cancer from benign diseases, respectively.

In a slightly different context, *Lee et al. (2013)* have developed a sandwich enzyme-linked assay that uses two different lectins that both bind to *O*-glycan structures of glycoproteins (Fig. 2C). The assay, which uses CGB lectin as capturing coated reagent and enzyme-conjugated jacalin as detection probe, was primarily designed to measure the levels of mucin-type *O*-glycosylated proteins in serum samples. When the assay was applied on sera of patients with stage 0 and stage I breast cancer as well as those of normal control women, significantly higher levels of *O*-glycosylated proteins were detected in both groups of breast cancer patients (*Lee et al., 2016*). The specificity and sensitivity of the assay were further improved when the same serum samples were subjected to perchloric acid enrichment prior to the analysis. Further characterization of the perchloric acid isolates by gel-based proteomics detected significant altered levels of plasma protease C1 inhibitor and proteoglycan 4 in both stage 0 and stage I breast cancer patients compared to the controls. Their data suggests that the ratio of the serum glycoproteins may be used for screening of early breast cancer.

## LECTIN HISTOCHEMISTRY

Like immunohistochemistry, lectin histochemistry is a microscopy-based technique for visualization of cellular components of tissues except that it uses lectin instead of antibodies. Utilization of labelled lectins in the tissue staining procedure limits the technique to detection of only glycan-conjugated components, as well as those whose glycan moieties are being recognized specifically by the individual lectins. Unlike immunohistochemistry which detects presence of specific antigens based on the specificities of antibodies used, lectin histochemistry provides information concerning glycosylation processes within a tissue sample as well as their intracellular locations. This information can be very useful in the characterization and/or detection of diseases.

In lectin histochemistry, labelling can be performed directly or indirectly (*Roth, 2011*). In the direct labelled method, which is generally less sensitive than the direct method, lectins are directly linked to fluorophores, enzymes, colloidal gold or ferritin, depending on the microscopy involved (Fig. 3A). On the other hand, the indirect method involves conjugation of lectins with biotin or digoxigenin, which may be detected using enzyme linked-streptavidin or -anti-digoxigenin, respectively (Fig. 3B). Apparently, not all chemicals can be used in the fixation and embedding of tissues in lectin histochemistry. For example, the use of formaldehyde in fixation of tissue specimens is known to cause reduced sensitivity of the *Griffonia simplicifolia* agglutinin, whilst ethanol-acetic acid fixation improved its binding (*Kuhlmann & Peschke, 1984*). Paraffin, which causes denaturation of proteins, is also known to result in attenuated binding of lectins due to sequestration of carbohydrates in the glycoproteins that are denatured. However, this can be largely reversed by removal of tissue-embedded paraffin using xylene or by trypsinization, which breaks the protein cross-links and allows the lectins to bind more efficiently (*Brooks & Hall, 2012*).

Lectin histochemistry has been extensively used in the study of glycosylation changes in cancer tissues. Two lectins have been found useful in distinguishing the different histological

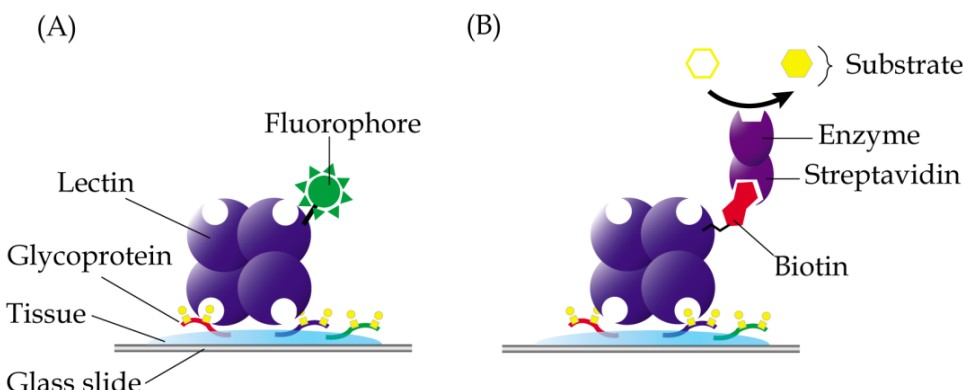

**Figure 3 Common techniques in lectin histochemistry.** Comparative staining of cancer versus normal tissues may highlight aberrant glycosylation of glycoproteins. (A) In the direct method, glycoproteins are detected in tissue specimens using a lectin that is covalently linked to fluorophores, enzymes, colloidal gold or ferritin. (B) The indirect labelled method, which is generally more sensitive, involves use of a lectin that is conjugated with a hapten, such as biotin or digoxigenin, which are then recognized using enzyme linked-streptavidin or -anti-digoxigenin, respectively.

grades of mucoepidermoid carcinoma, the most common type of salivary gland cancer (*Sobral et al., 2010*). Whilst ConA was demonstrated to be able to stain all grades of mucoepidermoid carcinoma tissues, staining with UEA-I lectin showed direct correlation of malignancy with the intensity of staining. Another example is cholangiocarcinoma which is attributed to the river fluke infection that commonly occurs in Thailand. In the study of the parasite-induced cancer, *Indramanee et al. (2012)* have used multiple lectins to demonstrate aberrant glycosylation of glycoconjugates in paraffin-embedded liver tissues of patients with primary cholangiocarcinoma. Unique lectin staining patterns derived from the cancer patients, relative to non-tumorous tissues, can be utilized as early stage markers for the bile duct cancer. Similarly, SNA has been proposed for use as a prognostic probe for invasive ductal carcinoma based on the different staining patterns that were generated compared to tissue sections of patients with stage 0 breast cancer, ductal carcinoma *in situ* (*Dos-Santos et al., 2014*). In another histochemical study, eight different lectins have been used to identify specific carbohydrates that may contribute to the progression of colorectal cancer (*Hagerbaumer et al., 2015*). The results showed changes in the binding patterns of five of the lectins during advancement of metastasis from adenoma to colorectal carcinoma.

## LECTIN BLOTTING

Lectin blotting is an extension of western blotting that uses lectin instead of antibody to detect glycoconjugates (*Shan, Tanaka & Shoyama, 2001*). As in western blotting, samples are similarly resolved using polyacrylamide gel electrophoresis and transferred onto a polyvinylidene fluoride (PVDF) or nitrocellulose membrane but detected using glycan-specific lectin probes (Fig. 4). Like histochemistry, visualization of the lectin complex is enabled via the use of conjugates such as enzymes, fluorescent dyes, biotin, digoxigenin, colloidal gold and radioactive isotopes. In lectin blotting, the concentrations of lectins used

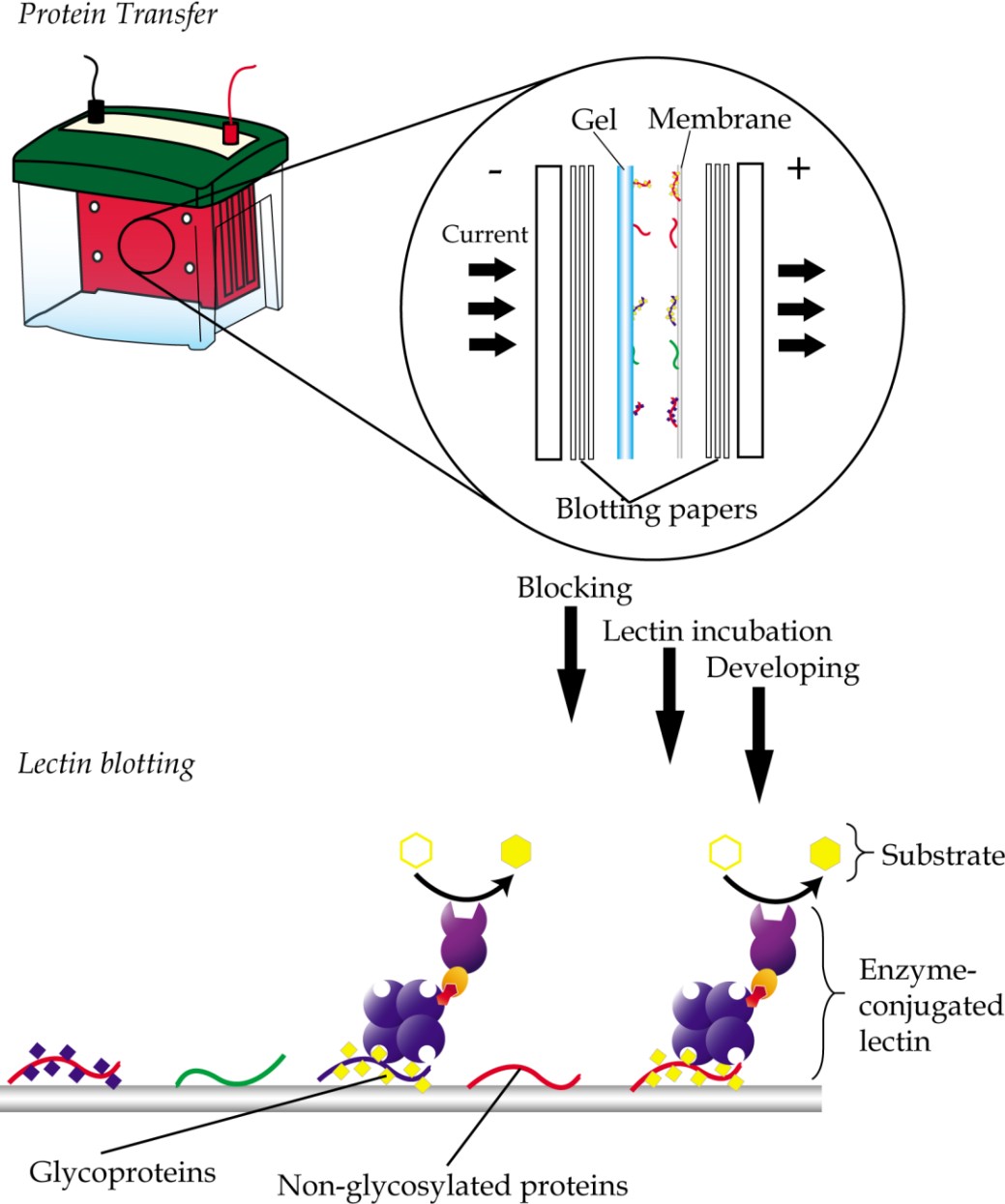

**Figure 4 General workflow of lectin blotting.** The method initially involves transferring of proteins that are resolved by gel electrophoresis onto a PVDF or nitrocellulose membrane. This is then followed by subjecting the membrane to washing, blocking and incubation with lectins that are conjugated to an enzyme, a fluorescent dye, biotin, digoxigenin, colloidal gold or radioactive isotopes. Comparative blotting of bodily fluids of cancer patients versus those from cancer negative subjects may highlight presence of aberrantly glycosylated and/or expressed glycoproteins.

must be at optimal levels to reduce false-positive binding. Although a powerful tool, this technique is however not quite suitable for routine diagnostics.

In the past, lectin blotting studies have been especially useful in characterization of structures of glycans (*Akama & Fukuda, 2006*), detection and quantification of *N*-

and *O*-glycosylated proteins (*Roth, Yehezkel & Khalaila, 2012*) and detection of altered glycosylation following an abnormality in glycosylation pathways due to disease processes (*Kitamura et al., 2003*). In cancer biomarker studies, lectin blotting is often used for comprehensive profiling of glycosylated proteins in biofluids. For example, the CGB lectin has been extensively used to demonstrate altered abundances of various *O*-glycosylated proteins in serum and/or urine samples of cancer patients that were resolved by 2-DE and transferred onto nitrocellulose membrane. Cancers that have been investigated using the method include endometrial cancer, cervical cancer (*Abdul-Rahman, Lim & Hashim, 2007*), breast cancer, nasopharyngeal carcinoma, bone cancer (*Mohamed et al., 2008*), ovarian cancer (*Mu et al., 2012*) and prostate cancer (*Jayapalan et al., 2012*; *Jayapalan et al., 2013*). Similar lectin blotting studies have also been applied on cell lines. Examples are the use of *Pinellia pedatisecta* agglutinin-based lectin blotting analysis to generate unique glycosylation fingerprints for leukemia and solid tumor cell lines (*Li et al., 2014*), and the utilization of ConA and CGB lectin to demonstrate altered released of *N*- and *O*-glycosylated proteins from murine 4T1 mammary carcinoma cell line (*Phang et al., 2016*).

Another use of lectin blotting is as a means of validation of tumor-specific glycosylation. Based on earlier results that showed elevated levels of mRNA of specific glycosyltransferases in endometroid ovarian cancer tissue relative to normal ovary, *Abbott et al. (2010)* have selected three different lectins (*Phaseolus vulgaris* erythroagglutinin, *Aleuria aurantia* lectin and *Datura stramonium* lectin) with distinctive affinities for the respective products of the enzymes to validate glycosylation changes of glycoproteins that are expressed in the ovarian cancer tissues. By extracting intact glycoproteins from the ovarian tissues before isolating the lectin-reactive proteins, the researchers were able to identify a total of 47 potential tumor-specific lectin-reactive markers. In another study, *Qiu et al. (2008)*, using biotinylated AAL and SNA lectin-blot detection method, were able to validate the differential *N*-linked glycan patterns that are related to the levels of sialylation and fucosylation of complement C3 in colorectal cancer patients, compared to those with adenoma and normal subjects. Similarly, *Park et al. (2012)* have validated earlier findings of aberration of fucose residues in haptoglobin β chain that is associated with progression of colon cancer by generating comparable results using *Lotus tetragonolobus* and *Aspergillus oryzae* lectins as detection probes in lectin blotting experiments.

## LECTIN ARRAY

Lectin array is a technique that was developed for rapid and sensitive analysis of glycans in a high-throughput manner. The technique uses multiple lectins, which are mostly plant-derived, that are immobilized onto a solid support at a high spatial density to detect different carbohydrate content of glycoproteins or glycolipids in a single sample (*Hu & Wong, 2009*; *Hirabayashi, Kuno & Tateno, 2011*). Display of the lectins in an array format enables observation of the distinct binding interactions simultaneously, which then provides a unique method for rapid characterization of carbohydrates on glycoconjugates (Fig. 5A). A glass slide is the most common material used as solid support for the array application. Lectins are coated on the glass surface either by covalent interaction or physical

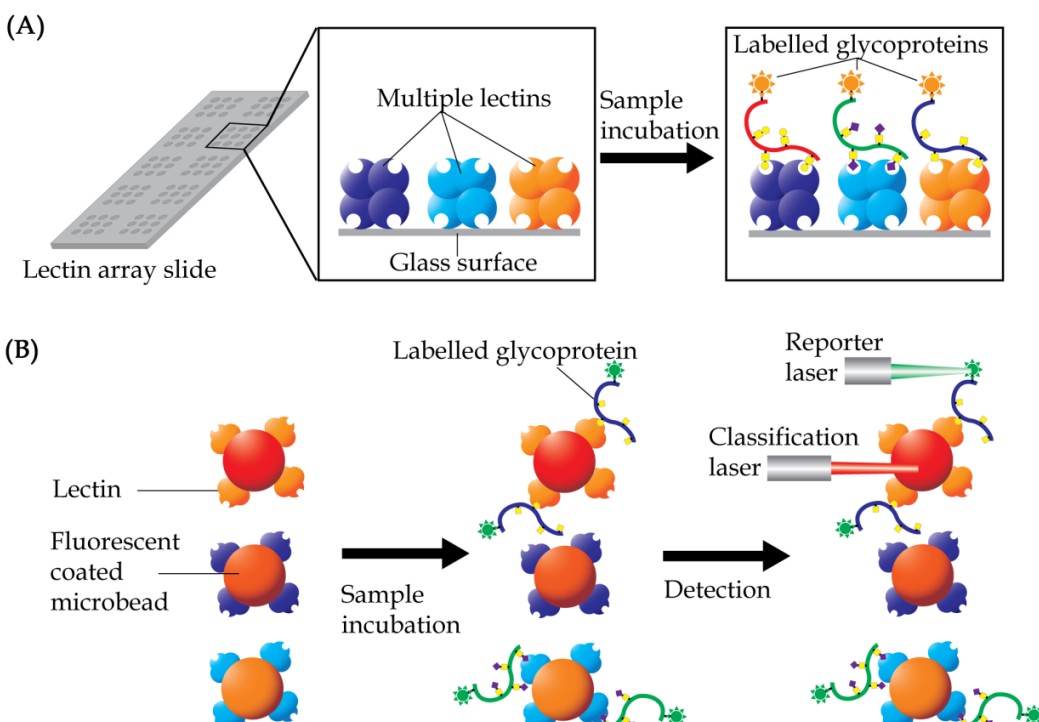

**Figure 5** **Basic concept of lectin array technology.** (A) Multiple lectins are printed onto a slide, which is organized in a grid, single lectin per spot, format. Samples, which are usually pre-labelled with either fluorophore or chromophore, are then allowed to interact with the lectins. Lectin spots, which contain the labelled glycoproteins, will illuminate under an appropriate scanner. (B) In lectin bead array analysis, different fluorescent colored beads, each corresponding to a single lectin, are often used. The conjugated beads are then allowed to interact with samples and the unbound materials being washed out. The beads are then passed through a detector with two laser sources, with the classification laser identifying the specific beads, whilst the reporter laser quantifies the presence of the labelled samples.

adsorption. Glass slides are usually pre-treated with chemical derivatives such as $N$-hydroxy succinimidyl esters (*Hsu & Mahal, 2006*), epoxides (*Kuno et al., 2005*), biotin, streptavidin (*Angeloni et al., 2005*), and 3D hydrogels (*Charles et al., 2004*). Each droplet of lectin is printed onto the glass slide and arranged according to a specific grid map using an array printer. The printed slide is held in place by a multi-well gasket, which allows samples to be loaded into each well.

By using an array of 45 different lectins to determine predictive biomarkers of colorectal cancer, *Nakajima et al. (2015)* were able to identify 12 lectins that showed increase binding, whilst 11 more lectins demonstrated low binding of glycoproteins in the colorectal cancer tissues compared to normal epithelia. Amongst the lectins, *Agaricus bisporus* lectin which was selected for further validation by the researchers, showed strong potential to be used as a new predictive biomarker for distant recurrence of curatively resected colorectal cancer. A similar approach performed on tissue extracts of gastric cancer demonstrated high interactions of 13 lectins with tissue glycoproteins, whilst 11 others showed low interaction (*Futsukaichi et al., 2015*). In both these studies, the altered interaction of lectins only

reflected the general presence of glycoproteins that were differently glycosylated without providing any information on the precise glycoproteins that are affected.

In an earlier study, *Wu et al. (2012)* have used lectin array to screen for altered fucosylated proteins in serum samples of patients with ovarian cancer. Based on the results, the researchers then immobilized the lectins that showed differential interactions and used it as affinity chromatography to isolate serum glycoproteins with aberrant glycan structures and determine their protein identities. This strategy has led to the identification of four serum glycoproteins with altered fucose residues. Recently, a different lectin array strategy was also developed to serve as an analytical technique for determination of differences in glycosylation of proteins that are isolated from serum samples (*Sunderic et al., 2016*). In this study, the glycan content of serum alpha-2-macrogobulin, which was isolated from serum samples of patients with colorectal cancer, was studied using the lectin array. From a set of 14 fluorescent labelled lectins that were used in the analysis, statistically significant differences between two groups of patients with colorectal cancer and cancer negative individuals were found for five of the lectins. When taken together, the results generally showed that the alpha-2-macrogobulin of patients with colorectal cancer have higher content of α2,6 sialic acid, GlcNAc and mannose residues, and tri-/tetraantennary complex type high-mannose $N$-glycans.

Since its inception, the technology of lectin array has been through several modifications to improve detectability of glycoproteins in biological samples. The array may involve prior pre-capturing of a glycoprotein of interest using antibody, and the subsequent detection of glycans using pre-labelled lectins (*Kuno et al., 2011*; *Li et al., 2011*). This approach allows detection of the total glycan content of a specific glycoprotein and also reduces the need for prior glycoprotein purification. Lectin array is not limited to glass slide as its solid support. *Wang et al. (2014)* have used fluorescent dyes coated microbeads, which allows multiplex detection in a single reaction vessel that greatly improves detection sensitivity compared to the standard lectin arrays. More recently, an alternative approach which involves printing of purified samples onto a chip surface has also been reported (*Sunderic et al., 2016*).

Lectin array analysis can also be performed on magnetic beads (Fig. 5B). Known as lectin magnetic bead array, the technique was first introduced as a robust and high-throughput pipeline for glycoproteomics-biomarker discovery in 2010 (*Loo, Jones & Hill, 2010*). The method is based on use of multiple lectins that are conjugated to magnetic beads to isolate glycan specific proteins. These lectin-conjugated beads are incubated with protein samples, washed and the bound glycoproteins are then eluted in appropriate buffers for subsequent proteomics analysis. By coupling a mass spectrometer to the one-step glycoprotein separation and isolation procedure, profiling of glycan-specific proteins may be achieved without much loss of proteins. This increases the probability of identification of proteins of lower abundances that have biomarker potentials. Nevertheless, a few methodological concerns need to be carefully considered when using the lectin bead array. These include surface functionality and diameter of the beads, conditions of buffers and duration of trypsin digestion protocols for optimal isolation of lectin-binding proteins. In this technique, understanding of the specificities of lectins is also imperative as most

glycosylated proteins are expected to have multiple glycosylation sites for interaction with the lectins.

Using a panel of 20 lectins in a magnetic bead array that was coupled to a tandem mass spectrometer, *Shah et al. (2015)* have demonstrated unique lectin-glycoprotein interactions in serum samples that may be used to distinguish three groups of subjects comprising healthy volunteers, patients with Barrett's esophagus and patients with esophageal adenocarcinoma. Their results demonstrated the possibility of using apolipoprotein B-100 to distinguish healthy volunteers from patients with Barrett's esophagus. The use of *Narcissus pseudonarcissus* lectin in the assay was able to differentiate differently glycosylated apolipoprotein B-100 in the two groups of subjects. On the other hand, patients with Barrett's esophagus were markedly distinguishable from those with esophageal adenocarcinoma via differences in the glycosylation of AAL-reactive complement component C9, whilst PHA-reactive gelsolin was shown to have potential in differentiating healthy subjects from patients with esophageal adenocarcinoma.

## CHALLENGES AND FUTURE DIRECTIONS

Development and progression of cancer are associated with altered glycosylation and aberrantly expressed glycoproteins. Hence, the use of lectin-based assays and strategies that are discussed in this review article, together with the emergence of proteomics technology, has led to identification of hundreds of putative glycopeptide biomarkers that can be utilized in clinical practice. A summary on the advantages and disadvantages of these lectin-based techniques is shown in Table 4. However, the translation of biomarkers from discovery to clinically approved tests is still much to be desired. This is mainly attributed to the lack of follow-up characterization and validation investigations of the potential biomarkers, which is an absolute requirement to ensure that the discovery phase experiments are not flawed and that detection of the biomarkers is reproducible, specific and sensitive (*Diamandis, 2012*; *Drucker & Krapfenbauer, 2013*). A potential glycopeptide biomarker has to be validated using hundreds of specimens to become clinically approved tests. Hence, this is certainly not possible in cases of rare cancers.

In some cases, validation may not be successful with the use of a single cancer biomarker in a single assay. One solution is to explore the simultaneously use of several different biomarkers for development of a highly specific and sensitive assay (*Pang et al., 2010*). Hence, there is an urgent need to consolidate data on the availability of all putative glycopeptide biomarkers that have been unmasked from the discovery phase studies for every different application in every cancer. In addition, new high throughput assays for simultaneous detection of multiple biomarkers are also required. The recent technological advances in chip-based protein microarray technology (*Sauer, 2017*) may provide with the solution, and therefore ought to be explored for simultaneous validation analysis of the different biomarkers in a single experiment.

In many other cases, identification of the potential glycopeptide biomarkers using lectin-based strategies may involve complex separation techniques such as 2-DE, which is laborious and expensive for large scale validation studies. 2-DE comes with the advantage

**Table 4  Advantages and disadvantages of lectin-based techniques in cancer biomarker discovery research.**

| Techniques | Advantages | Disadvantages |
|---|---|---|
| Lectin affinity chromatography | • Does not require purified glycoproteins or glycans<br>• Detailed analysis of glycan<br>• High affinity | • Requires large amounts of samples<br>• Time-consuming<br>• Allows for individual samples only<br>• Co-elution of other proteins |
| Enzyme-linked lectin assay (ELLA) | • Relatively high-throughput<br>• Quantitative<br>• Easy to perform<br>• Very cost effective<br>• Requires minute amounts of samples<br>• In case of hybrid ELLA, prior purification of a glycoprotein is not required | • Glycoproteins that are detected may not be identifiable unless it is coupled with further proteomics analysis or antibody detection.<br>• In case of hybrid ELLA, non-specific direct interaction of lectin to antibodies may occur<br>• Require purified glycans or glycoproteins as standard |
| Lectin histochemistry | • Simple<br>• Rapid<br>• Allows lectin multiplexing with the use of fluorescent tags | • Requires skills for tissue preparation<br>• Requires use of multiple lectins/antibodies to provide further confirmation<br>• Certain fixatives or components may reduce sensitivity |
| Lectin blotting | • Visualization of small amounts of proteins<br>• Easy to detect<br>• High specificity and sensitivity<br>• Reliable and reproducible<br>• Convenient method of screening of complex protein samples | • Choice of membrane may affect protein binding capacity and chemical stability |
| Lectin array | • Does not require purified glycoproteins or glycans<br>• Rapid<br>• Highly sensitive<br>• High-throughput<br>• Allows multiplexing<br>• Requires small amounts of samples | • Requires extensive optimization<br>• Possible non-specific interaction |

of knowing the actual experimental molecular weight of a glycopeptide biomarker, which is not possibly attained from liquid-based separation methods. This is important as many tumor associated glycopeptides are known to be truncated products of native glycoproteins (*Pinho & Reis, 2015*). For these potential biomarkers, validation experiments would need to involve a different indirect high-throughput technique using both lectin as well as an antibody that is capable of differentiating truncated glycopeptides from their native glycoprotein structures. However, such antibodies are usually not available commercially, and generating them is time consuming, costly and involves substantial laboratory work.

### Funding

This work was funded by FRGS-2015-1(FP0032015A) and HIR-MOHE H-20001-00-E000009 research grants from the Ministry of Higher Education, Malaysia. The funders had no role in study design, data collection and analysis, decision to publish, or preparation of the manuscript.

## Grant Disclosures

The following grant information was disclosed by the authors:

Ministry of Higher Education: FRGS-2015-1(FP0032015A), HIR-MOHE H-20001-00-E000009.

## Competing Interests

The authors declare there are no competing interests.

## Author Contributions

- Onn Haji Hashim conceived and designed the experiments, analyzed the data, wrote the paper, reviewed drafts of the paper.
- Jaime Jacqueline Jayapalan analyzed the data, wrote the paper, prepared figures and/or tables, reviewed drafts of the paper.
- Cheng-Siang Lee analyzed the data, wrote the paper, prepared figures and/or tables.

## Data Availability

The research in this article did not generate any data or code (literature review).

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
