# Peer review of "Lectins: an effective tool for screening of potential cancer biomarkers"

_PeerJ, doi:10.7717/peerj.3784_

## Round 0.1 · original submission · Minor Revisions

This review appears to be timely and is not a matter of review in the recent past. It is therefore of importance and of value to the readers of PeerJ. However, as you will note, there are a few concerns that need to be addressed before the paper is considered suitable for publication.

The most important is to break the review into subheadings so that the reader is able to see clearly what is being talked of in a particular section. In addition, a single table where the use of lectins is summarized, citing references, would be of great help to the reader. Finally, I strongly advise you to carefully go through the language in the article, since both reviewers have commented on some textual errors. Please take the help of a native English speaker, if required, to improve the article.

Reviewer 1 ·

Basic reporting

The literature review submitted by Onn Hashim and coworkers, is an exhaustive essay on the use of lectins as biomarkers in the diagnosis, classification and prognosis in oncological contexts. The review describes in rich details giving copious examples, the use of lectins using affinity chromatography, enzyme linked assays, cyto- and histo-chemistry, and blotting. The review ends with a section describing present and future challenges in lectin-based discoveries of potential biomarkers.
The review is well-written, lucid and will be of benefit to researchers in the field. A second positive feature is the extensive and updated list of references since most other reviews in this field tend to cite older papers, thoroughly understandable given that lectins have been used to interrogate cancers for about a century or so.
One important section that is missing from the review is a section on the use of fluorophore-conjugated lectins in cytometric investigation of glycochemical changes in cancer. Along with lectin histo- and cyto-chemistry, lectin chemistry when combined with flow cytometry can reveal potential intratumoral heterogeneities and in a more quantifiable manner. It is therefore surprising that this technique and associated body of literature on it is not represented in the essay.
The review would immensely benefit from a table which summarizes the specific uses, advantages and disadvantages of using each of the four techniques described in the manuscript.
There are a few minor grammatical errors which have been pointed in the manuscript version attached.

Experimental design

Being a literature review, there are no experiments performed, and hence, no experimental design

Validity of the findings

Being a literature review, there are no experiments performed, and hence, no findings to validate

Annotated reviews are not available for download in order to protect the identity of reviewers who chose to remain anonymous.

Reviewer 2 ·

Basic reporting

1. The manuscript should be carefully re-read for any English or grammar errors, especially in sentence formation. For e.g. Later, they are also classified based on the characteristics on page 2 of the manuscript.
2. Since it is a review, it might be a good idea to format it as subheadings throughout the manuscript, rather than introduction. Some suggested changed might be to add subheadings such as Lectin biology and biomarkers in cancer therapy for the sections within the introduction. Further, since the major body of the manuscript discusses the methods used to detect lectin / glycoproteins, a suitable heading might be used as well.

Experimental design

As mentioned in the previous section,
Since it is a review, it might be a good idea to format it as subheadings throughout the manuscript, rather than introduction. Some suggested changed might be to add subheadings such as Lectin biology and biomarkers in cancer therapy for the sections within the introduction. Further, since the major body of the manuscript discusses the methods used to detect lectin / glycoproteins, a suitable heading might be used as well.

Validity of the findings

1. While the authors have briefly talked about the biomarkers currently used in cancer research, it might be a good idea to briefly discuss the non- blood or non-secreted biomarkers that are used such as tissue based methods.
2. For the last section, it might be better to call it as challenges and future directions.

Additional comments

The article "Lectins: an effective tool for screening of potential cancer biomarkers" is a comprehensive review of the use of lectins and the methods by which it can be detected for use as cancer biomarkers. As mentioned in the above three categories, textual and formatting changes will make the article easier to read and follow. Further, a more comprehensive introduction on biomarkers currently used in cancer research, lectin or non-lectin based could be added before the discussion of the methods to increase the importance of the content discussed.

---

## Round 0.2 · accepted · Accept

Congratulations! Your paper is acceptable for publication.

Reviewer 1 ·

Basic reporting

The current revised manuscript meets the criteria.

Experimental design

N/A

Validity of the findings

N/A